Increased expression of POLR3G predicts poor prognosis in transitional cell carcinoma

Liu Xianhui 1
Zhang Weiyu 1 2
Wang Huanrui 1 2
Lai Chin-Hui 1
Xu Kexin cavinx@yeah.net 1
Hu Hao huhao@bjmu.edu.cn 1
1 Department of Urology, Peking University People’s Hospital , Beijing , China
2 Peking University Applied Lithotripsy Institute, Peking University People’s Hospital , Beijing , China
Verger Alexis
Electronic publication date: 2020 Nov 3
Publication date: 2020
Volume: 8
Electronic Location ID: e10281
Received 2020 Jul 10; Accepted 2020 Oct 9
Copyright: ©2020 Liu et al.
Copyright year: 2020
Copyright holder: Liu et al.
License: This is an open access article distributed under the terms of the Creative Commons Attribution License, which permits unrestricted use, distribution, reproduction and adaptation in any medium and for any purpose provided that it is properly attributed. For attribution, the original author(s), title, publication source (PeerJ) and either DOI or URL of the article must be cited.
License URL: https://creativecommons.org/licenses/by/4.0/

Keywords: Transitional cell carcinoma, POLR3G, Biomarker, TCGA, Tumor immue infiltration, Prognosis

Funding: The Research and Development Foundation of Peking University People’s Hospital RDX2018-04 Beijing Medical Award Foundation 2119000380 This work was supported by the Research and Development Foundation of Peking University People’s Hospital (No. RDX2018-04) and Beijing Medical Award Foundation (No. 2119000380). The funders had no role in study design, data collection and analysis, decision to publish, or preparation of the manuscript.

==============================
Background

Previous studies have shown that RNA Polymerase III Subunit G (POLR3G) has oncogenic effects in cultured cells and mice. However, the role of POLR3G in transitional cell carcinoma (TCC) has not been reported. This study explores the potential of POLR3G as a novel molecular marker for TCC.

Methods

The RNA sequencing data and clinical information of patients with TCC were downloaded from The Cancer Genome Atlas official website. Transcriptome analysis was performed as implemented in the edgeR package to explore whether POLR3G was up-regulated in TCC tissues compared to normal bladder tissues. The expression of POLR3G in bladder cancer cell line T24 and human uroepithelial cell line SV-HUC-1 were detected via quantitative real time polymerase chain reaction (qRT-PCR). Correlations between POLR3G expression and clinicopathological characteristics were analyzed using Mann-Whitney U test or Kruskal-Wallis H test. Clinicopathological characteristics associated with overall survival were explored using the Kaplan-Meier method and Cox regression analyses. Gene set enrichment analysis (GSEA) was performed to explore the associated gene sets enriched in different POLR3G expression phenotypes and the online tool Tumor IMmune Estimation Resource (TIMER) was used to explore the correlation between POLR3G expression and tumor immune infiltration in TCC.

Results

Transcriptome analysis showed that POLR3G was significantly up-regulated in TCC tissues compared to normal bladder tissues. Furthermore, qRT-PCR revealed high expression of POLR3G in T24 cells compared to SV-HUC-1 cells. Overall, POLR3G expression was associated with race, tumor status, tumor subtype, T classification, and pathological stage. Kaplan-Meier survival analysis revealed that higher POLR3G expression was associated with lower overall survival. The univariate Cox regression model revealed that age at diagnosis, pathological stage, and POLR3G expression were associated with prognosis of TCC patients. Further multivariate analyses identified these three clinicopathological characteristics as independent prognostic factors for overall survival. GSEA analysis showed that several gene sets associated with tumor development and metastasis, including TGF-β signaling, PI3K-AKT-mTOR signaling, and IL6-JAK-STAT3 signaling, were significantly enriched in POLR3G high expression phenotype. Immune infiltration analysis revealed that the expression of POLR3G was significantly correlated with infiltrating levels of immune cells, including CD8+ T cells, neutrophils, and dendritic cells; and the expression of POLR3G was also significantly correlated with the expression of immune checkpoint molecules, such as PD1, PD-L1, PD-L2, CTLA4, LAG3, HAVCR2, and TIGIT.

Conclusions

POLR3G was up-regulated in TCC and high POLR3G expression correlated with poor prognosis. POLR3G can potentially be used as a prognostic marker for TCC and might be of great value in predicting the response to immunotherapy.

Introduction

Bladder cancer is the 7th most commonly diagnosed cancer in males and the 11th most commonly diagnosed cancer when both genders are considered (Ferlay et al., 2013). In 2020, an estimated 81,400 new cases of bladder cancer (62,100 men; 19,300 women) will be diagnosed in the United States of America and approximately 17,980 deaths (13,050 men; 4,930 women) will occur during the same period of time (Siegel, Miller & Jemal, 2020). Transitional cell carcinoma (TCC) is the most common histological type of bladder cancer, contributing to more than 90% of all bladder cancer cases (Witjes et al., 2020). Approximately 75% of patients with bladder cancer are present with non-muscle-invasive bladder cancer (NMIBC) at the initial diagnosis, while the remaining 25% of patients are present with muscle-invasive bladder cancer (MIBC) or metastatic bladder cancer (Witjes et al., 2020). The standard treatment for NMIBC is trans-urethral resection of bladder tumor (TURBT) followed by intravesical chemotherapy or bacillus Calmette-Guérin (BCG) immunotherapy depending on risk stratifications. For MIBC, on the other hand, neoadjuvant chemotherapies followed by radical cystectomy are first line recommendations. However, the prognosis for MIBC is poor even with effective treatments. The five-year recurrence-free survival rate was 89% for patients with T2 tumors, 50% for patients with T4 tumors, and 35% for patients with lymph node metastasis respectively (Stein et al., 2001). Patients with NMIBCs have better prognoses, however, high grade NMIBC has a 70% recurrence rate with a 15%–40% risk of progression after five years (Kashif Khan, Ahmed & Raza, 2014).

For TCC, the most important histopathological prognostic variables are tumor stage and lymph node status (Dutta et al., 2016; Stein et al., 2001). However, no predictive molecular markers are routinely used in clinical practice. Thus, identifying effective markers is essential for predicting prognoses and directing treatments for patients with TCC. Although previous studies have revealed that RNA Polymerase III Subunit G (POLR3G) overexpression can have oncogenic effects in cultured cells and mice (Haurie et al., 2010; Khattar et al., 2016), its role in TCC has still not been reported. Herein, the aim of this study is to evaluate the correlation between POLR3G and prognoses of patients with TCC.

Materials & Methods

Data acquisition

The RNA sequencing (RNA-seq) data (Workflow types: HTSeq-FPKM; HTSeq-Counts) and corresponding clinical information of patients with TCC (Project: TCGA-BLCA, Disease type: transitional cell papillomas and carcinomas) were downloaded from GDC Data Portal (https://portal.gdc.cancer.gov/). Counts data were used for transcriptome analysis. TPM data were calculated based on FPKM data, and were used for further analysis. Only patients with both RNA-seq data and survival information were included in this study.

Cell lines and cell culture

Human bladder cancer cell line T24 and immortalized normal urothelial cell line SV-HUC-1 were purchased from National Infrastructure of Cell Line Resource (Beijing, China). Cell lines were maintained in Roswell Park Memorial Institute (RPMI) 1640 medium (Gibco) supplemented with 10% fetal bovine serum (Gibco) and 1% penicillin/streptomycin (Gibco). All cells were maintained in a humidified atmosphere with 5% CO2 at 37 °C.

RNA extraction and qRT-PCR

Total RNA was extracted from cells using the RNA simple Total RNA Kit (Tiangen). FastQuant RT Kit (Tiangen) was used for cDNA synthesis. The quantitative real time polymerase chain reactions (qRT-PCR) were performed using KAPA SYBR FAST Universal q-PCR Kit (KAPA). The relative mRNA levels of genes were calculated using cycle threshold (CT) methods, and β-actin was used as an endogenous control. Three replicate samples were studied for detection of mRNA expression. The primers were listed below: POLR3G (forward): 5′-CGCAGGCAAAGGCACAC-3′; POLR3G (reverse): 5′-CCTCTTTTTTCCAA TTCCTCCA-3′; β-actin (forward): 5′-CCAACCGCGAGAAGATGA-3′; β-actin (reverse): 5′-CCAGAGGCGTACAGGGATAG-3′.

Gene set enrichment analysis

Gene set enrichment analysis (GSEA) is widely applied to determine whether predefined gene sets are differentially expressed in different phenotypes (Subramanian et al., 2005). To identify signaling pathways that are differentially activated in TCCs, we conducted GSEA analysis between high and low POLR3G expression groups. GSEA analysis was performed using GSEA software (version 4.0.3). The h.all.v7.1.symbols.gmt (hallmark) dataset was obtained from the Molecular Signatures Database (MsigDB) (Liberzon et al., 2015). Enrichment analysis was performed by default weighted enrichment statistics, with the random combinatorial count set as 1,000. Gene sets were judged as significantly enriched by P < 0.05 as well as false discovery rates (FDR) < 0.25.

Tumor infiltrating immune cells and immune checkpoint molecule expression analysis

Tumor IMmune Estimation Resource (TIMER) (Li et al., 2017) is a web server for comprehensive analysis of tumor-infiltrating immune cells, which applies a previously published statistical approach called the deconvolution that uses the gene expression profiles to produce an inference on the number of tumor-infiltrating immune cells (Li et al., 2016). Survival module was used to explore the association between immune cell infiltration (B cells, CD4+ T cells, CD8+ T cells, neutrophils, macrophages, and dendritic cells) and clinical outcome in bladder cancer. Gene module was used to explore the correlations between the expression of POLR3G and immune cell infiltration. Correlation module was used to explore the correlations between POLR3G expression and immune checkpoint molecule expression, including PDCD1 (also known as PD1), CD274 (also known as PD-L1), PDCD1LG2 (also known as PD-L2), CTLA4, LAG3, HAVCR2, and TIGIT.

Statistical analysis

The data processing and further statistical analyses were performed using R (v.3.6.3)

Transcriptome analysis of the differentially expressed genes (DEGs) in the TCC tissues and normal bladder tissues was performed using edgeR package (Robinson, McCarthy & Smyth, 2010). The differential expression of POLR3G between T24 cells and SV-HUC-1 cells was analyzed via independent t test. The relationships between clinicopathological characteristics and POLR3G expression were analyzed using the Mann–Whitney U test or Kruskal-Wallis H test. The Kaplan–Meier method and Cox regression models were used to explore the influence of POLR3G expression on overall survival along with other clinicopathological characteristics (age at diagnosis, gender, race, and pathological stage). The cut-off value of POLR3G expression was determined by its median value. P < 0.05 was considered to indicate a statistically significant difference.

Results

Clinical characteristics

The clinical characteristics of 404 patients with TCC in The Cancer Genome Atlas (TCGA) are presented in Table 1. There were 299 (74.01%) male patients and 105 (25.99%) female patients with a median age of 69 years old at the time of diagnosis. Race information was available for 388 patients, among which 322 (82.99%) patients were white, 43 (11.08%) were Asian, and 23 (5.93%) were black or African American. At least 202 (63.72%) patients were tumor free while 115 (36.28%) patients still had tumors. Overall, 2 (0.50%) of the patients showed stage I, 129 (32.09%) stage II, 137 (34.08%) stage III, and 134 (33.33%) stage IV. Among 363 patients diagnosed with clear N stage, 129 (35.54%) had lymph node metastases. From 205 patients diagnosed with clear M stage, 11 (5.37%) had distant metastases. Median follow-up for subjects alive at last contact was 21.3 months (range 0–169 months).

Table 1 Patient characteristics of patients with TCC in TCGA.

Clinical characteristics	Total (n = 404)	Percentage (%)	
Age at diagnosis (y)			
≤60	86	21.34	
>60	317	78.66	
Gender			
Female	105	25.99	
Male	299	74.01	
Race			
White	322	82.99	
Asian	43	11.08	
Black or African American	23	5.93	
Tumor status			
With tumor	115	36.28	
Tumor free	202	63.72	
Tumor subtype			
Papillary	110	32.93	
Non-papillary	224	67.07	
T classification			
T1	3	0.81	
T2	119	32.08	
T3	192	51.75	
T4	57	15.36	
Lymph node metastasis			
Negative	234	64.46	
Positive	129	35.54	
Distant metastasis			
Negative	194	94.63	
Positive	11	5.37	
Pathological stage			
Stage I	2	0.50	
Stage II	129	32.09	
Stage III	137	34.08	
Stage IV	134	33.33	
Notes.

TCC Transitional cell carcinoma

TCGA The Cancer Genome Atlas

POLR3G was up-regulated in multiple cancer types including TCC

TIMER analyses revealed that POLR3G expression was up-regulated in multiple cancer types, including cholangiocarcinoma, colon adenocarcinoma, esophageal carcinoma, kidney renal clear cell carcinoma. (Fig. 1A). RNA-Seq differential expression analysis revealed that 2,211 genes were up-regulated (log2FC > 1, and FDR < 0.01) in TCC tissues compared to normal bladder tissues, including POLR3G (log2FC = 1.038, FDR = 0.006), and 1,853 genes were down-regulated (log2FC <-1, and FDR <0.01). (Fig. 1B and Table S1). Furthermore, qRT-PCR results showed higher expression levels of POLR3G in T24 cells compared to SV-HUC-1 cells (P = 0.004; Fig. 1C).

Figure 1 POLR3G expression in different disease states and survival curve of patients with TCC.

(A) Differential expressions of POLR3G in multiple cancer types analyzed by TIMER. P value significant codes: 0 ≤ *** < 0.001 ≤ ** < 0.01 ≤ * < 0.05 ≤ . < 0.1 (B) RNA-Seq differential expression analysis in TCC tissues and normal bladder tissues. (C) Differential expressions of POLR3G in T24 cells compared to SV-HUC-1 cells. (D) Impact of POLR3G expression on overall survival of patients with TCC in TCGA.

Figure 2 Association with POLR3G expression and clinicopathological characteristics.

Clinicopathologic characteristics included were listed as followed: (A) T classification. (B) Pathological stage. (C) Tumor status. (D) Tumor subtype. (E) Race. (F) Age at diagnosis. (G) Gender. (H) Lymph node metastasis. (I) Distant metastasis.

Relationship between POLR3G expression and clinicopathological characteristics

Expression levels of POLR3G were strongly correlated with T classification (T1-2 vs. T3-4, P = 0.001; Fig. 2A), pathological stage (stage I-II vs. stage III-IV, P = 0.005; Fig. 2B), tumor status (tumor free vs. with tumor, P = 0.001; Fig. 2C), tumor subtype (papillary vs. non-papillary, P < 0.001; Fig. 2D), and race (P < 0.001; Fig. 2E). No statistically significant differences were observed between groups stratified by age (≤ 60 years old vs. >60 years old, P = 0.113; Fig. 2F), gender (female vs. male, P = 0.072; Fig. 2G), lymph node metastasis (positive vs. negative, P = 0.201; Fig. 2H), and distant metastasis (positive vs. negative, P = 0.056; Fig. 2I).

Survival outcomes and Cox regression analysis

Kaplan–Meier survival analysis revealed that higher expression of POLR3G was associated with worse prognoses (P = 0.002; Fig. 1D). The univariate Cox regression model revealed that age at diagnosis (HR = 1.03, 95% CI = 1.02 to 1.05, P < 0.001), pathological stage (HR = 1.71, 95% CI = 1.41 to 2.08, P < 0.001), and POLR3G expression (HR = 1.04, 95% CI = 1.01 to 1.07, P = 0.02) were associated with overall survival of patients with TCC. Furthermore, multivariate Cox regression after adjustment indicated that age at diagnosis (HR = 1.03, 95% CI = 1.02 to 1.05, P < 0.001), pathological stage (HR = 1.77, 95% CI = 1.45 to 2.17, P < 0.001), and POLR3G expression (HR = 1.05, 95% CI = 1.02 to 1.08, P = 0.001) were independent prognostic factors for overall survival in patients with TCC (Table 2, Fig. 3).

Table 2 Univariate and multivariate Cox regression analysis of overall survival.

Characteristics (n = 381)	Univariate analysis	Multivariate analysis	
	HR (95% CI)	P value	HR (95% CI)	P value	
Age at diagnosis	1.03 (1.02–1.05)	<0.001	1.03 (1.02–1.05)	<0.001	
Gender	0.82 (0.59–1.16)	0.263	0.84 (0.60–1.18)	0.321	
Race	1.04 (0.78–1.38)	0.783	1.28 (0.97–1.68)	0.084	
Pathological stage	1.71 (1.41–2.08)	<0.001	1.77 (1.45–2.17)	<0.001	
POLR3G expression	1.04 (1.01–1.07)	0.021	1.05 (1.02–1.08)	0.001	
Notes.

HR hazard ratio

CI confidence interval

Figure 3 Multivariate Cox analysis of POLR3G expression and other clinicopathological characteristics.

Gene sets enriched in POLR3G high expression phenotype

In the hallmark dataset, 41 gene sets were significantly enriched in POLR3G high expression phenotype (Table 3). Several of the gene sets are associated with oncogenesis, progression, and metastasis of cancer, such as mitotic spindle, hypoxia, Kras signaling up, PI3K-AKT-mTOR signaling, IL6-JAK-STATS3 signaling, mTORC1 signaling, TNF- α signaling via NF- κB, inflammatory response, and Myc targets v1(Figs. 4A–4I).

Table 3 Gene sets enriched in POLR3G high expression phenotype.

Gene set	NES	NOM P value	FDR q value	
IL6 JAK STAT3 signaling	2.83	<0.001	<0.001	
Allograft rejection	2.72	<0.001	<0.001	
mTORC1 signaling	2.7	<0.001	<0.001	
Protein secretion	2.69	<0.001	<0.001	
Inflammatory response	2.67	<0.001	<0.001	
Glycolysis	2.58	<0.001	<0.001	
Heme metabolism	2.56	<0.001	<0.001	
Mitotic spindle	2.53	<0.001	<0.001	
Androgen response	2.46	<0.001	<0.001	
Interferon-γ response	2.46	<0.001	<0.001	
TNF-α signaling via NF-kB	2.44	<0.001	<0.001	
PI3K AKT mTOR signaling	2.42	<0.001	<0.001	
G2M checkpoint	2.36	<0.001	<0.001	
Hypoxia	2.34	<0.001	<0.001	
UV response down	2.32	<0.001	<0.001	
Apical junction	2.28	<0.001	<0.001	
Kras signaling up	2.28	<0.001	<0.001	
Complement	2.28	<0.001	<0.001	
Myc targets v1	2.22	<0.001	<0.001	
Unfolded protein response	2.18	<0.001	0.006	
UV response up	2.08	<0.001	0.011	
IL2 STAT5 signaling	2.06	<0.001	0.015	
Interferon-α response	2.03	<0.001	0.014	
Epithelial mesenchymal transition	2.03	<0.001	0.014	
E2F targets	2.03	<0.001	0.013	
Apoptosis	2.02	<0.001	0.013	
TGF-β signaling	2.02	<0.001	0.012	
Adipogenesis	2	<0.001	0.012	
Coagulation	1.96	<0.001	0.015	
Reactive oxygen species pathway	1.88	<0.001	0.026	
P53 pathway	1.87	<0.001	0.03	
Fatty acid metabolism	1.87	<0.001	0.029	
Estrogen response early	1.84	<0.001	0.028	
Myc targets v2	1.83	<0.001	0.027	
DNA repair	1.77	<0.001	0.035	
Oxidative phosphorylation	1.77	<0.001	0.034	
Angiogenesis	1.66	<0.001	0.056	
Apical surface	1.62	<0.001	0.057	
Spermatogenesis	1.59	<0.001	0.065	
Estrogen response late	1.57	<0.001	0.076	
Cholesterol homeostasis	1.46	<0.001	0.087	
Notes.

NES normalized enrichment score

NOM P value nominal P value

FDR false discovery rate

Figure 4 Enrichment plots from gene set enrichment analysis.

Gene sets enriched in POLR3G high phenotype: (A) Mitotic spindle. (B) Hypoxia. (C) Kras signaling up. (D) PI3K-AKT-mTOR signaling. (E) IL6-JAK-STATS3 signaling. (F) mTORC1 signaling. (G) TNF-α signaling via NF-κ B. (H) inflammation response. (I) Myc targets v1.

POLR3G expression was associated with levels of immune cell infiltration and immune checkpoint molecule expression

Analysis of TIMER survival module indicated that the infiltration of CD8+ T cells is related to the cumulative survival rate in TCC (P = 0.006, Fig. 5A). Gene module analysis revealed that POLR3G expression was negatively correlated with tumor purity (cor = −0.178, P = 6.02e −04) and positively correlated with infiltrating levels of CD8+ T cell (cor = 0.317, P = 5.70e −10), neutrophil cells (cor = 0.237, P = 5.12e −06), and dendritic cells (cor = 0.399, P = 2.15e−15) in TCC (Fig. 5B). Moreover, Correlation module analysis revealed that the expression of POLR3G was significantly correlated with the expression of immune checkpoint molecules including PDCD1 (cor = 0.227, P = 3.57e −06), CD274 (cor = 0.455, P = 3.18e −22), PDCD1LG2 (cor = 0.399, P = 4.60e−17), CTLA-4(cor = 0.279, P = 9.87e−09), LAG3 (cor = 0.344, P = 8.32e−13), HAVCR2 (cor = 0.312, P = 1.18e−10), and TIGIT (cor = 0.255, P = 1.78e−07) (Fig. 5C).

Figure 5 Correlation of POLR3G expression with immune cell infiltration and immune checkpoint molecule expression by TIMER.

(A) Correlation of immune cell infiltration with prognoses of patients with TCC. (B) Correlation of POLR3G expression with immune cell infiltration. (C) Correlation of POLR3G expression with immune checkpoint molecule expression.

Discussion

Several studies have investigated prognostic biomarkers for TCC, such as FGF2 (Shariat et al., 2010), UHRF1 (Unoki et al., 2009), and GRIA1 (Tilley, Kim & Fry, 2017). However, there are still no ideal predictive molecule for clinical application. This study demonstrates that POLR3G is a potentially useful biomarker for predicting prognosis of TCC.

POLR3G is an RNA polymerase III peripheral subunit that synthesizes small RNAs, such as 5S rRNA, tRNAs, and some microRNAs (Haurie et al., 2010). POLR3G plays a role in sensing and limiting infection by intracellular bacteria and DNA viruses, acts as a nuclear and cytosolic DNA sensor involved in innate immune responses, and is also essential for the maintenance of stem cell state (Ablasser et al., 2009; Chiu, Macmillan & Chen, 2009; Lund et al., 2017; Wong et al., 2011). Several studies have described the links between POLR3G and cancer. For example, Durrieu-Gaillard et al. reported that POLR3G expression was strongly up-regulated during the process of tumoral transformation in the human lung fibroblast cell line IMR90 model system (Durrieu-Gaillard et al., 2018). Haurie et al. showed that overexpression of POLR3G in IMR90 increased the expression of genes associated with tumor growth and metastasis, including S100A4, RFC2, EZR, and RAC1, and reduced the expression of tumor-suppressing genes, such as PFDN5 and KLF6 (Haurie et al., 2010). Another study (Petrie et al., 2019) found that the expression of POLR3G was up to three-fold higher in prostate tumors compared to normal adjacent samples. Similar results were observed at the cellular level; POLR3G expression was elevated in the prostate cancer cell line PC-3 compared to the immortalized healthy prostate epithelium cell line PNT2C2. In addition, knockdown of POLR3G triggered the proliferative arrest of PC-3.

Results of this study revealed that POLR3G was highly expressed in multiple cancer types, including TCC, and qRT-PCR further confirmed that POLR3G was elevated in T24 cells compared to SV-HUC-1 cells. These findings are indicative of a cumulative alteration of POLR3G expression during TCC tumorigenesis. The traditional perspective of TCC tumorigenesis postulates that TCCs arise via two different but overlapping pathways: papillary pathway and non-papillary pathway (Dinney et al., 2004). We found that non-papillary TCCs exhibited higher POLR3G expression compared to papillary TCCs. Therefore, POLR3G might play different roles in these two pathways. We also found that high POLR3G expression was positively correlated with high T classification, advanced clinical stage, and tumor recurrence, which are strongly correlated with poor prognosis in patients with TCC. More importantly, further univariate and multivariate analysis identified POLR3G expression as an independent prognostic factor for overall survival.

We conducted GSEA analysis to investigate the relationship between POLR3G and gene signatures in TCCs. Our results showed that 41 gene sets were significantly enriched in the POLR3G high expression group, including mitotic spindle, Inflammatory response, TGF-β signaling, epithelial mesenchymal transition, PI3K-AKT-mTOR signaling, and IL-6-JAK-STATS3 signaling. Several of these pathways are associated with oncogenesis, progression, and metastasis of cancer, suggesting that POLR3G expression contributes to the development, progression, and prognosis of TCC. However, the regulatory mechanism needs to be further elucidated.

Immunotherapy is a key treatment approach for TCC. Intravesical BCG immunotherapy has been used to treat superficial TCC for over 40 years and still represents the first-line adjuvant treatment for superficial TCC after TURBT to prevent tumor recurrence (Babjuk et al., 2019; Morales, Eidinger & Bruce, 1976). Over the past decade, immune checkpoint inhibitor (ICI) immunotherapy breakthroughs have enriched the available treatment modalities for advanced TCCs. Atezolizumab and Pembrolizumab have been approved for first-line systemic therapy for cisplatin-ineligible patients with local advanced or metastatic TCC whose tumors express PD-L1 (Balar et al., 2017a; Balar et al., 2017b). However, the objective response rate (ORR) to ICIs in bladder cancer patients was only ∼20% (Balar et al., 2017a; Balar et al., 2017b). Thus, identifying reliable biomarkers to distinguish which patients are more likely to respond to ICI immunotherapy is crucial for the successful treatment. Previous studies have demonstrated that the level of the immune infiltration within tumors correlates with bladder cancer prognosis and is a positive prognostic indicator of response to immunotherapy (Fridman et al., 2012; Pfannstiel et al., 2019). Furthermore, the expression of immune checkpoint PD-L1 on tumors correlates with unfavorable prognosis, but can also predict the immunotherapy reactivity of patients (Thompson, Dong & Kwon, 2007; Topalian, Drake & Pardoll, 2015). Therefore, we explored the potential role of POLR3G in immune cell infiltration within TCCs using TIMER.

First, we used the survival module to explore the association between immune infiltrate abundance and clinical outcome. Previous studies have reported that CD8+ T cells infiltration might play a positive role in the prognosis of colorectal cancer (Naito et al., 1998), triple-negative breast cancer (Vihervuori et al., 2019), and pancreatic cancer (Masugi et al., 2019). However, our results suggest that CD8+ T cell infiltration was negatively correlated with cumulative survival in TCC. Second, we used the gene module to explore the correlation between POLR3G expression and immune infiltrate abundance. Results from this analysis showed that POLR3G expression was significantly correlated with the level of infiltrating immune cells in TCC. More specifically, POLR3G expression was negatively correlated with tumor purity, and positively correlated with the infiltrating levels of CD8+ T cells, neutrophil cells, and dendritic cells in TCC. We further explored the correlations between POLR3G and immune checkpoint molecules in TCC via the correlation module. Results revealed that POLR3G expression was significantly correlated with several immune checkpoint molecules, including PDCD1, CD274, PDCD1LG2, CTLA4, LAG3, HAVCR2, and TIGIT. Taken together, these findings suggest that POLR3G contributes to the regulation of immune cell infiltration and immune checkpoint molecule expression, resulting in the suppression of anti-tumor immunity. These results provide a possible mechanistic explanation for the worse prognosis observed in patients with higher POLR3G expression.

To the best of our knowledge, this is the first study investigating the role of POLR3G in TCC

We found that POLR3G expression was an independent prognostic factor for overall survival and can potentially be used as a prognostic biomarker in TCC. However, there were some limitations to this study. First, this study was conducted using data from the public database TCGA, and the clinical information was incomplete for some patients. Further investigation with a larger sample size is needed to validate our findings. Second, the relationships between POLR3G and immune cell infiltration were analyzed using online tools, which need to be further elucidated via molecular experiments.

Conclusions

In summary, POLR3G expression was up-regulated in TCC and can potentially be used as a prognostic marker. In addition, the expression of POLR3G was associated with levels of immune cell infiltration and the expression of immune checkpoint molecules in TCC, suggesting potential value for predicting patient response to ICI immunotherapy.

Supplemental Information

Supplemental Information 1 The results of transcriptome analysis using edgeR

Click here for additional data file.

The results shown here are mostly based upon data generated by the TCGA Research Network: http://cancergenome.nih.gov/.

Additional Information and Declarations

Competing Interests

Author Contributions

Data Availability

The authors declare there are no competing interests.

Xianhui Liu conceived and designed the experiments, performed the experiments, analyzed the data, prepared figures and/or tables, authored or reviewed drafts of the paper, and approved the final draft.

Weiyu Zhang performed the experiments, analyzed the data, prepared figures and/or tables, authored or reviewed drafts of the paper, and approved the final draft.

Huanrui Wang and Chin-Hui Lai analyzed the data, authored or reviewed drafts of the paper, and approved the final draft.

Kexin Xu and Hao Hu conceived and designed the experiments, authored or reviewed drafts of the paper, and approved the final draft.

The following information was supplied regarding data availability:

Data analyzed in this study are available at TCGA:

https://portal.gdc.cancer.gov/repository?facetTab=cases&filters=%7B%22op%22%3A%22and%22%2C%22content%22%3A%5B%7B%22op%22%3A%22in%22%2C%22content%22%3A%7B%22field%22%3A%22cases.disease_type%22%2C%22value%22%3A%5B%22transitional%20cell%20papillomas%20and%20carcinomas%22%5D%7D%7D%2C%7B%22op%22%3A%22in%22%2C%22content%22%3A%7B%22field%22%3A%22cases.primary_site%22%2C%22value%22%3A%5B%22bladder%22%5D%7D%7D%2C%7B%22op%22%3A%22in%22%2C%22content%22%3A%7B%22field%22%3A%22cases.project.program.name%22%2C%22value%22%3A%5B%22TCGA%22%5D%7D%7D%5D%7D&searchTableTab=cases.

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
