# Peer review of "Increased expression of POLR3G predicts poor prognosis in transitional cell carcinoma"

_PeerJ, doi:10.7717/peerj.10281_

## Round 0.1 · original submission · Minor Revisions

Your paper has been seen by two expert reviewers and as you will see below, they are supportive of publication. They have raised some minor concerns that will need to be satisfactorily addressed. We will consider publishing your manuscript only if you can accommodate their suggestions in a revised version or explain satisfactorily why their comments are invalid.

We look forward to receiving your revised manuscript.

Reviewer 1 ·

Basic reporting

Overall manuscript is looking good, still need to improve the English.
Some sentences are unclear.

Experimental design

Overall experimental and statistical analysis is good.

Validity of the findings

no comment

Additional comments

The manuscript "Increased expression of POLR3G predicts poor prognosis in transitional cell carcinoma" is well written by Liu et al. I have few comments:
1) "Data acquisition" section is too short and unclear.
2) capitalize the "cell lines and cell culture ".
3) First sentence of the "Gene set enrichment analysis" is similarity with the GSEA-MSigDB GitHub, reframe the sentence.
4) "Statistical downloaded from TCGA were merged and conducted using R (v.3.6.3). "; sentence is unclear.
5) Why they used TPM in Figure 1a and FPKM in Figure 1c?
6) I didn't find Fig. 5. in the manuscript.
7) Need professional editing.

Reviewer 2 ·

Basic reporting

The structure of the manuscript is suitable and contains all necessary sections (Abstract, introduction, methods, results, discussion and references). Some raw data are shared, although the tables provided are not referred in the text, not labelled and potentially problematic (all "standard deviations" are zero?). The figures needs to be better labelled, with clear legends (for instance, explicit what is blue and what is red in Figure 1A.
Few English errors or typos: line 52-53 "may of great value" should be "may be of great value".
In multiple places, "There’re" should be replaced by "There are"

Experimental design

The authors of the manuscript investigated the expression level of POLR3G in the TCGA dataset of Transitional cell carcinoma. They identified systematic overexpression of the gene in tumors samples compared to normal samples and a potential association of POLR3G level with survival. They then explored the association of POLR3G expression with clinicopathologic characteristics, with gene sets (using GSEA) and with immune infiltration (using TIMER). The importance of POLR3G overexpression might be overstated here, because the test used may not be suitable for this context (DESeq2 or edgeR are typically better for dealing with count data coming from RNASeq experiments). The authors do not indicate how many genes would be "overexpressed" with the test they used, or how much overexpression there is in tumor samples. In addition, it is not surprising that many genes are differentially expressed in tumors compared to adjacent normal cells, because of all the molecular processes occuring in tumors and the different cell composition in the samples.

Validity of the findings

The results are presented with adequate support. Conclusions are derived from the results, and limitations of the study are indicated.

---

## Round 0.2 · accepted · Accept

I sent this back to the two original reviewers and as you will see, they both find that you have addressed all the issues raised.

Reviewer 1 ·

Basic reporting

Revised manuscript looking better than the previous one.

Experimental design

The experimental design is good. The statical design of the experiment is correct.

Validity of the findings

Statistical analysis is robust and reproducible. The overall analysis is good enough to establish the role of POLR3G in prognosis in transitional cell carcinoma.

Additional comments

The authors make changes that were suggested. In the revised manuscript, they add extra information and changed the figures.

Reviewer 2 ·

Basic reporting

The manuscript is suitable and well-written

Experimental design

The study is essentially a re-analysis of an existing dataset and is both relevant and meaningful

Validity of the findings

The conclusions are well supported and the limitations are well described

Additional comments

My comments were addressed by the authors and the manuscript is suitable for publication on PeerJ